# Health Risks of Polybrominated Diphenyl Ethers (PBDEs) and Metals at Informal Electronic Waste Recycling Sites

**DOI:** 10.3390/ijerph16060906

**Published:** 2019-03-13

**Authors:** Chimere May Ohajinwa, Peter M. van Bodegom, Oladele Osibanjo, Qing Xie, Jingwen Chen, Martina G. Vijver, Willie J. G. M. Peijnenburg

**Affiliations:** 1Institute of Environmental Sciences (CML), Leiden University, P.O. Box 9518, 2300 RA Leiden, The Netherlands; p.m.van.bodegom@cml.leidenuniv.nl (P.M.v.B.); vijver@cml.leidenuniv.nl (M.G.V.); 2Department of Chemistry, University of Ibadan, Ibadan 200284, Nigeria; oosibanjo@yahoo.com; 3Key Laboratory of Industrial Ecology and Environmental Engineering (Ministry of Education), School of Environmental Science and Technology, Dalian University of Technology, Dalian 116024, China; qingxie@dlut.edu.cn (Q.X.); jwchen@dlut.edu.cn (J.C.); 4Center for Safety of Substances and Products, National Institute of Public Health and the Environment (RIVM), P.O. Box 1, 3721 Bilthoven, The Netherlands

**Keywords:** electronic waste, informal recycling, PBDEs, metals, soil, dust

## Abstract

Concerns about the adverse public health consequences of informal electronic waste (*e*-waste) recycling are increasing. This study adopted a cross-sectional study design to gain insights into health risks (cancer and non-cancer risks) associated with exposure to *e*-waste chemicals among informal *e*-waste workers via three main routes: Dermal contact, ingestion, and inhalation. The *e*-waste chemicals (PBDE and metals) were measured in the dust and top soils at *e*-waste sites (burning, dismantling, and repair sites). Adverse health risks were calculated using the EPA model developed by the Environmental Protection Agency of the United States. The concentrations of the *e*-waste chemicals and the health risks at the *e*-waste sites increased as the intensity of the *e*-waste recycling activities increased: control sites < repair sites < dismantling sites < burning sites. Dermal contact was the main route of exposure while exposure via inhalation was negligible for both carcinogenic and non-carcinogenic risks. Cumulative health risks via all routes of exposure (inhalation, ingestion, and dermal contact) exceeded the acceptable limits of both non-cancer effects and cancer risk at all *e*-waste sites. This indicates that overall the *e*-waste workers are at the risk of adverse health effects. Therefore, the importance of occupational safety programs and management regulations for *e*-waste workers cannot be over emphasised.

## 1. Introduction

Information Communication Technology (ICT) has revolutionized our everyday life, consequently causing an increasing demand for ICT. This growing importance of ICT coupled with rising obsolescence due to rapid technological advancements, demand for the latest ICT, and decreasing electrical electronic equipment (EEE) lifetime has led to a rapid increase in the volume of electrical electronic equipment discarded, which is known as Waste Electrical Electronic Equipment (WEEE, also known as *e*-waste) generated around the globe. *e*-Waste consists of electrical and electronic devices including all separate components (such as wires, cables, batteries, and circuit boards), which are at the end of their useful life [1,2]. *e*-Waste is one of the most complex waste streams because of the wide variety of components, compositions, and rapidly changing product designs. It is also the fastest growing municipal waste streams in the world.

The global estimate of *e*-waste generated in 2014 was 41.8 million metric tons, which increased to 44.7 million metric tons in 2016, and 52 million metric tons are expected to be generated by 2021 [3]. Of the quantity generated, only about 20% of *e*-waste generated is properly collected and recycled. About 80% of the *e*-waste generated globally is recycled in informal settings in developing countries such as Nigeria, Ghana, Brazil, Mexico, China, India, Vietnam, and the Philippines [4,5]. The concern with *e*-waste is not only about the volumes generated but also about the unsafe methods employed in recycling the electronics in developing countries, known as informal recycling. Informal *e*-waste recycling is unregulated, unorganised and often overlooked [6,7,8]. It perpetuates due to a lack of infrastructure for sound *e*-waste management, lax environmental laws/regulations, and weak enforcement of existing national and international laws/regulations [9,10,11,12], such as the implementation of extended producer responsibility schemes by manufacturers, which is already enforced in developed countries [2,3]. Informal *e*-waste recycling involves the use of crude methods such as the undocumented collection of *e*-waste from homes, workshops, dumpsites, sorting, manual dismantling, smelting, and open burning. These activities are carried out without safety precautions. This leads to the release of hazardous mixture chemicals into the environment. These practices have both environmental and public health consequences. However, information on the potential cancer and non-cancer risks associated with informal *e*-waste recycling is scarcely available in developing countries. Therefore, abating the public health implications of unsafe e-waste recycling practices could be a challenge without adequate information.

*e*-Waste contains a wide range of substances, some of which are economically valuable, and some are hazardous. Some of the hazardous substances are compounds of potential concern (COPC), which include metals, products of incomplete combustion (PICs), and/or reformation products. PICs include any organic compound emitted during incomplete combustion, whereas reformation products are organic compounds that are formed immediately after combustion, due to the interaction of specific constituents. Some of the organic compounds are persistent organic pollutants (POPs) such as brominated flame retardants (BFRs) like Polybrominated Diphenyl Ethers (PBDEs), non-dioxin like Polychlorinated Biphenyls (PCBs), Polycyclic Aromatic Hydrocarbons (PAHs), Polychlorinated Dibenzo-p-dioxins, and Furans (PCDD/Fs). These POPs, along with other organic compounds, may pose significant implications for human health and environmental safety [6,7,13,14].

In this study, we considered PBDEs as a proxy for the cocktail of POPs emitted at informal *e*-waste recycling sites. POPs like PBDEs are toxic, highly persistent in the environment, bio-accumulate in food chains, and they have a high potential for long-range environmental transport. In addition, metals from *e*-waste are non-biodegradable, they persist in the environment and can disturb the ecological balance of the aquatic and the terrestrial environment, as well as affect human health. These chemicals have been detected in humans and in increasing concentrations in various environmental matrixes, including air, water, soil, sediment, animals, and foods in all regions of the world [15]. Evidence of effects of exposure to informal *e*-waste recycling include injuries [8,16], infection of wounds, skin and eye injuries and irritations, respiratory problems [17,18], and noise pollution, occupational stress, among others [19]. There is also evidence on harmful effects of long-term exposure of humans and wildlife, including effects on fetal/child development, impacts on thyroid and neurologic functions, immunotoxicity, reproductive toxicity, and endocrine disruption with endpoints related to induction of cancer [17]: See Table 1 for more information on health effects due to exposure to organic and metal contaminants.

High concentrations of metals and PBDEs were found at and around informal *e*-waste recycling sites [35,36,37,38,39,40]. Large quantities of *e*-waste are informally recycled in Nigeria using various recycling activities such as repair, dismantling, and open burning [41]. Each of these activities may pose a potential significant source of human exposure to pollutants (toxic metals and organic pollutants). Human exposure could be through direct inhalation, ingestion, dermal contact, or via consumption of contaminated food and water. Thus far, to our knowledge, no study has estimated the health risks associated with informal *e*-waste recycling. Therefore, there is a need to estimate the health risks associated with exposure to *e*-waste chemicals such as metals and PBDEs. The most evident health-related issues are associated with direct occupational exposure. In addition to these apparent occupational acute health issues, there might be some unforeseen threatening health issues in the long run or even after the person has stopped working at *e*-waste sites. To provide an understanding of the health risks to which various informal *e*-waste workers in Nigeria are exposed to, it was hypothesized that the top soils and dusts samples from different informal *e*-waste sites may generate different levels of risks depending on the pollutant concentrations.

Therefore, the objectives of this study were to assess non-cancer and cancer risks that are attributable to metals and PBDEs in soils and dusts from different *e*-waste recycling sites. We estimated the health risks of exposure to metals and PBDEs pollution, as present in top soils (0–10 cm) and various dust samples (floor dust, and direct dust from electronics). We did this by calculating average daily doses for workers exposed via inhalation, dermal contact, and oral ingestion. We consider exposure to PBDEs and metals as a proxy for organic and inorganic chemicals respectively. Informal *e*-waste workers are inadvertently exposed to both classes of chemicals at the same time. In this paper we evaluated17 PBDE congeners: BDE-17, BDE-28, BDE-71, BDE-47, BDE-66, BDE-100, BDE-99, BDE-85, BDE-154, BDE-138, BDE-183, BDE-190, BDE-208, BDE-206, and BDE-209, as well as 24 metals Ag, As, Ba, Cd, Cr, Co, Cu, Fe, Ga, Ge, Hg, Mn, Ni, Pb, Se, Sn, Sb, Te, Ti, Ta, V, and Zn, at the various sites.

## 2. Methods

### 2.1. Study Locations and Designs

The methods employed in this study have been well detailed in our previous studies [8,16,35,36]. In brief, a cross-sectional study design was adopted to gain an understanding of the pollution levels at the *e*-waste recycling sites in the three study locationsin Nigeria: Ibadan, Lagos, and Aba. In each study location, a multi-stage random systematic sampling technique was used to select the sites. This was to ensure representative inclusion of various *e*-waste recycling activities (burning, dismantling, and repair) in the selected *e*-waste recycling areas. In Lagos, the selected sites were the Computer village, Ikeja (6.593° N, 3.342° E), and Alaba international market Ojor (6.462° N, 3.191° E). In Ibadan, the selected sites were Ogunpa (7.383° N, 3.887° E) and Queens Cinema areas (7.392° N, 3.883° E). In Aba, the shopping center(5.105° N, 7.369° E) and Port-Harcourt Road/Cementary (5.104° N, 7.362° E) and Jubilee road/St Michael’s Road (5.122° N, 7.379° E) were selected (Figure 1). Soil and dust samples were collected from the selected sites depending on the feasibility of collecting such samples. For metal analysis, a total of 62 samples were collected from all the sites. The 62 samples consist of 23 top soil (0–10 cm depth), 31 floor dust, three roadside dust, and five direct dust samples collected from the inside and outside of electronic devices were analyzed. For the PBDE analysis, a total of 56 samples consisting of 16 top soils (0–10 cm), 29 floor dust, 5 roadside dust, and 6 direct dust samples collected from the inside and outside of electronic devices were analyzed: See Appendix A. The difference in the number of samples analyzed for metals and PBDE is because there was loss of samples, and some samples were below the detection limits due to strong matrix effects. See the supplementary information for more details on the methods used for the analysis of metals and PBDEs.

### 2.2. Description of Recycling Activities and Likely Exposure Pathways

The recycling activities include collection, sorting, storage, washing, cleaning, dismantling, and metal recovery through stripping of wires or open burning. Most *e*-waste recycling activities (especially at dismantling and burning sites) are carried out outdoors, which involve manual dismantling (disassembling) using hammer, machetes, or any tool that can help separate the parts. Open burning leads to incomplete combustion and processed materials from the various *e*-waste activities are dumped outside on bare ground (no vegetative cover on the ground). Most repair activities, which involve soldering of various parts, take place indoors, but also sometimes outdoors, depending on the settings of the work environment and the weather condition. These activities release large quantities of hazardous substances without any emission control.

The workers work with minimal or no health or the environmental protection. The majority (82%) of the workers work without the use of any personal protective equipment (PPE) such as gloves, nose mask. Also most of them work in shorts, short-sleeved shirts, and slippers, exposing most parts of their body [8,16]; also see Appendix A (photos of *e*-waste workers at the sites). This means that they have multiple routes of exposure (directly and indirectly) to the *e*-waste chemicals. The exposure routes are via ingestion, inhalation, or dermal contact. Informal *e*-waste recycling happens mostly in urban slums, usually with no official governance, regulations, and people work mainly for economic benefits. Within the *e*-waste recycling vicinities, there are other (non-*e*-waste recycling) informal businesses with workers having similar socio-demographic information like the *e*-waste workers. In some locations there are water bodies less than 2 km away from the burning sites. In addition, most residences use boreholes (ground water) and deep wells as a source of water, as confirmed by Healya et al., 2017 [41]. Historically, from the responses of the *e*-waste workers and residents around the *e*-waste recycling sites, *e*-waste recycling activities seem to be the most critical activity that releases hazardous substances in the vicinity. Due to stricter enforcement of the *e*-waste regulations by the National Environmental Standards and Regulations Enforcement Agency (NESREA), Nigeria, the *e*-waste dumpsites/burning sites have been moved more than once at Alaba, Lagos. After a while the new sites were crowded with both old and new in-coming workers (usually migrants from northern Nigeria in search of greener pasture in the cities). As the migrants settle around the dumpsites, the sites finally turn into small temporary unplanned residential communities. One major concern is that current *e*-waste sites could be used for other activities in the future, which means that the impact of the emissions from *e*-waste recycling could go beyond the *e*-waste workers. We recognize that children around the *e*-waste recycling sites may be exposed to *e*-waste mixture chemicals, but in this study, we focus on *e*-waste workers’ exposure to metals and PBDEs that are likely to be emitted during *e*-waste recycling.

### 2.3. Health Risk Assessment

The potential health risks was assessed as the likelihood of adverse health effects resulting from exposure of *e*-waste workers to *e*-waste chemicals (metals and PBDEs) over a specified time period. The risks are commonly expressed in terms of the exceedance of the average daily dose (ADD). The ADD is based on the magnitude, frequency, and duration of human exposure to chemicals (PBDEs and metals in this study) in the environment. Information on the socio-demographic (age, weight, height) and occupational characteristics were obtained from the *e*-waste workers, which were used for the health risk estimates. The health risk of each of the metals, each of the PBDE congeners, and ∑PBDEs is expressed in terms of either carcinogenic risks or non-carcinogenic health hazards. Exposure to PBDEs and metals can occur via three main pathways: (a) Direct inhalation of vapor or of atmospheric particulates through mouth and nose; (b) incidental ingestion of dust and top soils due to their deposition on food or drinks or via hand-to-mouth activity, and (c) dermal absorption of substances present in particles adhering to exposed skin [42]. The models used in this study to calculate the exposure of humans to metals and PBDEs in dust and soil is based on the models developed by the Environmental Protection Agency of the United States [43,44,45,46].

The average daily dose (ADD) (mg/kg/day) of a pollutant in soil and dust taken up via ingestion, dermal contact, and inhalation, as exposure pathways wereestimated using Equations (1)–(3). ADD_ingestion_, ADD_inhalation_, and ADD_dermal_ are the daily amounts of PBDEs and metals taken up through ingestion, inhalation, and dermal contact (mg/kg/day), respectively. Median concentrations of the pollutants were used in these calculations. Table 2 presents the sources of the values and factors used for the health risk estimations and the meanings of the abbreviations.
(1)$$ADD_{ingestion}=C\times \frac{R_{ing}\times EF\times ED}{BW\times AT}\times CF$$
(2)$$ADD_{dermal}=C\times \frac{SA\times AF\times ABS\times EF\times ED}{BW\times AT}\times CF$$
(3)$$ADD_{inhalation}=\frac{C_{dust}\times R_{inh}\times ET\times EF\times ED}{PEF\times BW\times AT}\times CF$$

Based on the ADDs, and the toxicity risk indices, the health risks for non-cancer hazards and cancer risks) of the PBDEs and metals were estimated using Equations(4) and (5). The Hazard Quotient (HQ) is used to calculate the non-carcinogenic risks based on reference daily dose (RfD or reference concentrations (RfC) [50]. The RfD is the toxicity value used in evaluating the adverse health effects; is an estimate of the allowable daily exposure to the human population [49]. Values for RfD and RfC were available only for four PBDEs congeners (BDE-47, BDE-99, BDE-153, and BDE-209), and for 19 metals: Ag, As, Ba, Cd, Cr, Co, Cu, Fe, Hg, Mn, Ni, Pb, Se, Sn, Sb, Ti, Ta, V, and Zn from USEPA 2011 [48] and USEPA 2017 [50]. For further estimation of the potential cancer risks, only seven metals (Cr, Co, Ni, As, Cd, Hg, and Pb) and one PBDE congener (BDE-209) toxicity values available [49]. A HQ value below one indicates that there is an acceptable level of risk (indicating low probability of any adverse effect), while HQ values exceeding one are indicative of unacceptable risks (higher than acceptable probability of an adverse health effect). HQ values exceeding one are assumed to be of concern [50]. The HQ for each of the pollutants (PBDEs and metals) was calculated for ingestion, dermal contact, and inhalation pathways, respectively.

Thecarcinogenic risk is the probability of an individual developing any type of cancer from the lifetime exposure to carcinogenic chemicals. The health risk for carcinogen risk characterization is based on the slope factor (SF) or the Inhalation Unit Risk (IUR). The slope factor (mg kg^−1^ day^−1^) is used in risk assessment to estimate the lifetime probability of an individual developing cancer as a result of exposure to a particular carcinogen. A risk above 1 × 10^−4^ is generally considered to be unacceptable, a risk below 1 × 10^−6^ is considered not to trigger any health effect, while risks calculated to be in between 1 × 10^−4^ and 1 × 10^−6^ are within the acceptable limits. A risk of 1 ×10^−6^ is interpreted as indicating that an individual has a one in 1,000,000 chance of developing cancer from the exposure evaluated by [51,52,53].
Oral Hazard Quotient (HQ_ing_) = ADD/RfD
Inhalation Hazard Quotient (HQ_inh_) = ADD/RfC
Dermal Hazard Quotient (HQ_der_) = ADD/(RfD × GIABS)
(4)
Carcinogenic risk_ing_ = ADD_ing_ × SF
Carcinogenic risk_inh_ = ADD_inh_ × IUR
Carcinogenic risk_der_ = ADD_der_ × (SF × GIABS)(5)

GIABS is the gastrointestinal absorption factor which was assumed to be equal to one (assuming the total absorption of contaminants for all congeners) [48].

In addition, the Hazard Index (HI) is used to assess the potential of exposure to multiple chemicals or multiple exposure routes at the sites to cause non-carcinogenic effects through different pathways. The hazard index is equal to the sum of the HQ values for the individual chemicals. Since the workers are exposed to multiple substances (both metals and PBDEs) within individual exposure pathways at the same time, we estimated the total non-cancer hazard by summing up the HIs of metals and PBDEs for each of the exposure routes [49]. We assume there are no interactions between PBDEs and metals.

### 2.4. Ethical Considerations

Ethical approval was obtained from the University of Ibadan/University College Hospital Ethical Review Board (No. UI/EC/15/0096). Verbal and written consent of the workers was obtained at the start of the interview, after explaining to the workers their full rights to refuse and to withdraw at any time during the interview. To ensure that the participant remains anonymous each questionnaire was coded with number identifiers. They were also assured that the data will not be used for other purposes than for scientific reasons and for the development of safety promotion programs for the sector. Permission to conduct the study was also obtained from the association of second-hand electronics dealers at each study site. This study is a part of a bigger study.

## 3. Results

### 3.1. Descriptive Statistics of the PBDE and Metals

In the Appendix A, a summary of the medians of the various PBDE and metal concentrations in soil and in dust samples at the various *e*-waste recycling sites is shown for each of the three study locations. The general pattern of the PBDEs and metal distribution in top soil and dust samples from the sites showed concentrations in this increasing order: Control sites < repair sites < dismantling sites < burning sites. The concentrations of most of the PBDE and metals congeners at the *e*-waste sites in the three locations exceeded the concentrations at the corresponding control sites.

### 3.2. Human Health Risk Assessment

#### 3.2.1. Quantitative Estimation of Non-Carcinogenic Effects

The HI values for dermal exposure to metals and PBDEs combined, via soil and dust were greater than one at all *e*-waste recycling sites (burning, dismantling, and repair sites), with metals contributing the great majority of the risk.. This indicates that the concentrations at those sites exceeded the threshold (safe) limit and the workers are at risk of developing non-cancer health effects via dermal exposure, followed by ingestion of soil and dusts at the sites. In contrast, the non-carcinogenic risks via inhalation were negligible. Dermal contact was shown to be the main route of exposure to both metals and PBDEs and consequently poses a higher risk. Generally for the fourPBDEs, BDE-209 contributed most to the health risk, followed by BDE-99 in Lagos and Ibadan, while Aba BDE-99 contributed the most, followed by BDE-153. Figure 2 and Figure 3 present the hazard index (HI) of top soils and dust for non-cancer risks via all exposure pathways at the *e*-waste sites for the three locations for PBDEs and metals, respectively. See Appendix A for more details, Appendix A shows the RfD, RfC, and GIABS used for the estimates.

Combining the HIs for non-cancer risks of metals and PBDEs also revealed that the total HI exceeded the acceptable (safe) limit for non-cancer hazards via dermal exposure at all sites in all locations, and via ingestion of direct dust at repair sites in Ibadan and ingestion of top soils at burning sites in Lagos (Figure 4 and Appendix A). The total HI is mainly influenced by the HI of metals. We also considered the cumulative non-cancer effects by summing the risks from all exposure routes. The cumulative non-cancer effects exceeded the accepatable limit at all sites in all locations, see Figure 5.

#### 3.2.2. Quantitative Estimation of Risk of Developing Cancers

The cancer risk for BDE-209 via ingestion is within the range of 2.3 × 10^−13^ to 4.72 × 10^−9^, and for dermal uptake is between 3.2 × 10^−10^ to 6.6 ×10^−6^. This indicates that the risks of developing cancer via dermal contact, since the ranges are above the safe limit of 1 × 10^−6^ especially at the burning sites; see Figure 6 and Appendix A for more details. These findings indicate that exposure of *e*-waste workers to PBDEs is potentially harmful to their health.

Only seven metals (Cr, Co, Ni, As, Cd, Hg, and Pb) had toxicity values available forthe estimation of potential cancer risks via ingestion, inhalation, and dermal contact. The HI for cancer risk through metals via ingestion ranged from 8.7 × 10^−6^ to 1.5 × 10^−4^, via the inhalation it ranged from 9.1 × 10^−15^ to 1.5 × 10^−14^, andvia dermal contact it ranged from 7.0 × 10^−4^ to 1.2 × 10^−1^, see Figure 7. For more details on the results for each of the locations, see Appendix A. These results show that exposure via inhalation induces risks that are below the acceptable (safe) limit, while exposure via ingestion and dermal contact induces risks that exceeded the acceptable (safe) limits at all sites in all locations. These findings indicate that the workers are most at risk of adverse non-cancer health effects via dermal contact, followed by ingestion, while exposure via inhalation induces negligible risks; dermal > ingestion > inhalation routes. Burning sites seem to be the most unsafe sites followed by dismantling and repair sites. These findings indicate that exposure of *e*-waste workers to metals is harmful to their health.

Since the workers are exposed to multiple substances (both metals and PBDEs) at the same time within individual exposure pathway, we estimated the total cancer risk by summing up the HIs of metals and PBDEs for each of the exposure routes. The total HI is mostly influenced by the HI of metals. The total HI shows that the exposure via ingestion and dermal contact of metals and PBDEs exceeded the acceptable (safe) limit for cancer risks at all sites in all locations (Figure 8 and Appendix A). We also considered it appropriate to sum risks from multiple exposure routes (i.e the cumulative risk of exposure) (See Reference [49], Exhibit 8–1 to 8–3). Obviously, the cumulative cancer risk also exceeded the acceptable (safe) limit at all sites in all locations, see Figure 9 and Appendix A.

## 4. Discussion

As far as we are aware, this is one of the few studies that estimated the cancer risks and non-cancer hazards of exposure to PBDEs and metals in soil and dust samples from different informal *e*-waste activity sites (burning, dismantling, and repair sites). The strength of this study is that we considered three exposure pathways, different *e*-waste recycling activities, various environmental samples (top soils and dusts) from different types of *e*-waste recycling and compared exposure in three different cities in two different geopolitical zones in Nigeria. We estimated non-cancer effects, cancer risks, and the assessed the cumulative effect of exposure to both PBDEs and metals via all exposure routes. We also used some primary data on exposure parameters obtained from the respondents for the risk estimation, instead of using default US EPA exposure parameters, which are commonly used in other studies. In addition, we used epidemiological methods to select the target groups and sites to ensure that the results obtained are a representative of the target groups and sites, and that the findings are applicable to similar situations anywhere in the world. Focus on the three types of informal *e*-waste recycling activities provided a comprehensive insight on the health risks for different groups of *e*-waste workers. Workers are exposed to far more chemicals than the ones considered here. Therefore, this study can be considered as being indicative of the risks due to both organic and inorganic chemicals. One limitation to this study is that there was no air sampling, which would have revealed deeper understanding on the exposure level via inhalation.

The findings of this study as performed in Nigeria are likely to be representative for informal *e*-waste recycling in developing countries that lack the resources for safe *e*-waste recycling. Increasing amounts of electronic waste, unsafe recycling methods and disposal pose significant risks to the environment and human health, therefore hindering sustained health. Understanding the implications of scientific data related to informal *e*-waste recycling contributes towards the achievement of Sustainable Development Goals (SDGs) related to environmental protection (Goals 6, 11, 12, and 14), health (Goal 3), and Goal 8 that focuses on employment and economic growth [54].

### 4.1. Health Risk Assessments

The concentrations of the PBDEs and metals considered in this study showed overall an increasing trend of health risks at the sites as the intensity of the *e*-waste activities increased in this order: Control sites < repair sites < dimantling sites < burning sites. This is similar to the findings of many studies reviewed by Ni et al., 2013 [55]. This finding reveals that open burning of *e*-waste is the most risky recycling activity. The risks associated with the high levels of *e*-waste chemicals and poor work practices call for concern.

The health risk assessment shows the impact of the different metals and PBDE congeners via viarious routes (ingestion, inhalation, and dermal contact). Overall, the magnitude of exposure to non-cancer effects and cancer risks of PBDEs via the various routes is in this order: Dermal contact, followed by ingestion of soil and dust at all the sites (burning, dismantling, and repair sites). The same pattern of exposure risks are revealed for metal exposure: Dermal contact, followed by ingestion, while exposure via inhalation is negligble (dermal contact > ingestion > inhalation). This finding is consistent with a similar study on *e*-waste sites reported by Civan and Kara 2016 [56] and review studies by Song et al., 2015 [57]. The health risk for the metals were much more pronouced than those for PBDEs.

The cumulative hazard index of PBDEs and metals via all exposure routes at all the *e*-waste sites exceeded the acceptable (safe) limits by several orders of magnitude. Considering the exposure via the different routes to both PBDEs and metals, the total risks calculated show that exposure via dermal contact exceeds the acceptable limit for non-cancer effects in all locations at all sites with the burning sites having the highest risks. Similarly, the total risks for ingestion and dermal contact exceeded the acceptable limits for cancer risks in all locations at all sites with the burning sites having the highest risks. For both non-cancer effects and cancer risks, dermal exposure is the main route of exposure. In addition, the cumulative health risks via all routes of exposure (inhalation, ingestion, and dermal contact) exceeded the acceptable limits of both non-cancer effects and cancer risk at all *e*-waste sites and in all locations.

### 4.2. Implications for Health Risks

The risk assessment indicates that overall the *e*-waste workers are at the risk of adverse health effects. Therefore, the occupational safety program for the *e*-waste workers, especially the use of personal protective equipment (PPE), cannot be over emphasised. PPE such asappropriate work cloths, as most of their body parts are exposed (see Appendix A). The majority (82%) of the *e*-waste workers do not use any PPE, and one the reasons for not using PPE is discomfort. The workers complained about the discomfort of using PPE at work, e.g., the gloves and thick coveralls because it hinders productivity; with the gloves feel hot after some time and is difficult to pick up tiny screws, and the coveralls are not convenient for the weather (hot and humid) in the tropics. This observation is in accordance with a study on preferred product characteristics for PPE in tropical climates by World Health Organization (WHO), in which the need for comfort is necessary for the increased use of PPE and safety. In addition, a study by de Almeida et al., 2012 [58] highlighted the need for thermal comfort of PPE to increase PPE usage by workers. The primary purpose of PPE is to prevent injury, but comfort is also important because it can dramatically influence whether workers actually make proper use of it.

Exposure of *e*-waste workers to PBDEs, metals, and other hazardous substances is even worse, because, surprisingly, 88% of the workers are unaware that *e*-waste contains hazardous chemicals, and 70% do not think that the chemicals in *e*-waste can pose any health risk. In the study by Ohajinwa et al., 2017a [8], the workers had a low health risk awareness level. This shows that informal workers often appear to underestimate or deny the health risks associated with their jobs. This could be because this job is a means of livelihood for them and they cannot escape from the risks easily. This is in accordance with the theory of Cognitive Dissonance proposed by Festinger in 1957 in which he stated that recognition of inconsistency will cause dissonance, and will motivate an individual to resolve the dissonance by either change of beliefs, change of actions, or change of perception of action [59].

Ohajinwa et al., 2017 also reported a positive correlation between workers’ knowledge and work practice. Therefore, improving *e*-waste workers’ knowledge on the health risks associated with their daily jobs may decrease risky practices [8]. It is crucial that *e*-waste workers are educated on the potential health risks peculiar to their jobs and the safety measures to be undertaken. It is also important to note that non-*e*-waste workers, residents, and children around the *e*-waste sites are very likely to also be at risk of adverse health effects from informal *e*-waste recycling as also pointed out by References [15,56,57,60]. The other informal non-*e*-waste workers around the *e*-waste recycling vicinity have similar socio-demographics and work conditions [61], hence they are likely to be exposed to similar health risks like the *e*-waste workers. The high metal and PBDE concentrations at the *e*-waste sites may also be an indirect source of pollution of surface and ground water and air, and could consequently affect people farther away from the *e*-waste sites. It should therefore be noted explicitly that the actual health risks would be higher than the risks calculated in this study. This indicates an urgent need for more appropriate and effective policies, regulations, and strategies for enforcement actions suitable for the informal sector.

We recommend:That government and other formal institutions design effective occupational health and safety (OSH) programs for the informal *e*-waste workers.The enforcement of the policies and regulations. One effective way to enforce safety is for formal institutions to work with the informal *e*-waste recycling associations to identify comfortable PPE, and to communicate the health risks peculiar to informal *e*-waste recycling and the safety measures to be undertaken. However, to effectively implement any OSH program in this sector, it should be borne in mind that the approach must be situation specific. The approach may differ depending on the type of job performed and location. In addition, enforcement agencies must not be seen to be at cross-purposes with the informal *e*-waste sector, as it frequently appears because informal associations have proved to operate efficiently without any formal support.The ban of open burning of *e*-waste and other risky practices. If open burning of *e*-waste is not banned, the effects will consequently affect those living far away from the recycling sites through pollution of soil, air, underground water, and contamination of plants and foods. These contaminants might even affect the unborn generation. One way to ensure such a ban is to (a) devise appropriate alternative ways of *e*-waste recycling with caution to protect health and environment, (b) bridge the communication gap between enforcement agencies and informal *e*-waste workers, and (c) for the informal *e*-waste recycling associations to be made accountable for safer practices.More studies on air monitoring of the *e*-waste recycling sites, especially at the burning sites as fine particles would not have been captured in the samples analyzed for this study. Air monitoring of the site might reveal exposure via inhalation as a significant route of exposure.The use of the hierarchical control method in the informal *e*-waste recycling sector(Figure 10). Such controls are simple steps that will help to minimize exposure and health risks associated with informal *e*-waste recycling, without impeding the workers’ source of livelihood. This will not only protect the *e*-waste workers, but also protect people around the *e*-waste recycing sites.

In our assessment, we assumed that there are no interactions of chemicals that increase toxic effects to humans. Moreover, it is known that the toxicity of the chemicals also depends on other parameters such as exposure time, dose, age, oxidation state, solubility, and properties of the environment among others [62]. To address these uncertainties, we recommend further studies on biomonitoring of informal *e*-waste workers. In addition, we could not assess exposure via inhalation of dust samples at burning sites. Using respirable particles (dust samples) at burning sites is needed in future studies. We recommend further toxicological studies to determine the cumulative toxicological effect associated with exposure to multiple chemicals through different exposure pathways, because the additive response method applied in this study might underestimate the potential for health effects. We also recognize that we did not identify all compounds present in the emissions at the *e*-waste recycling sites, including those in the new list of one of the most used intenational referencestandard guideline values (SGVs) for environmental pollutants [63].

## 5. Conclusions

Our study is one of few studies thatestimate the total non-cancer effects and cancer risksof *e*-waste chemicals (organic and inorganic) that *e*-waste workers and people around the *e*-waste recycling vicinity may be exposed to. The *e*-waste workers are prone to both adverse non-carcinogenic and carcinogenic health risks. The magnitude of exposure showed that dermal contact is the most important exposure route, followed by ingestion, while exposure via inhalation is the least important exposure route. This is even more worrisome as previous studies revealed that *e*-waste workers have poor work practices and low awareness of the health risks associated with their work. These sobering findings call for the need for urgent action by both national and international govenments. There is a need for more appropriate *e*-waste management regulations that consider maximum participation of the informal *e*-waste workers to ensure a moresustainable improvement and development in this sector.

## Figures and Tables

**Figure 1 ijerph-16-00906-f001:**
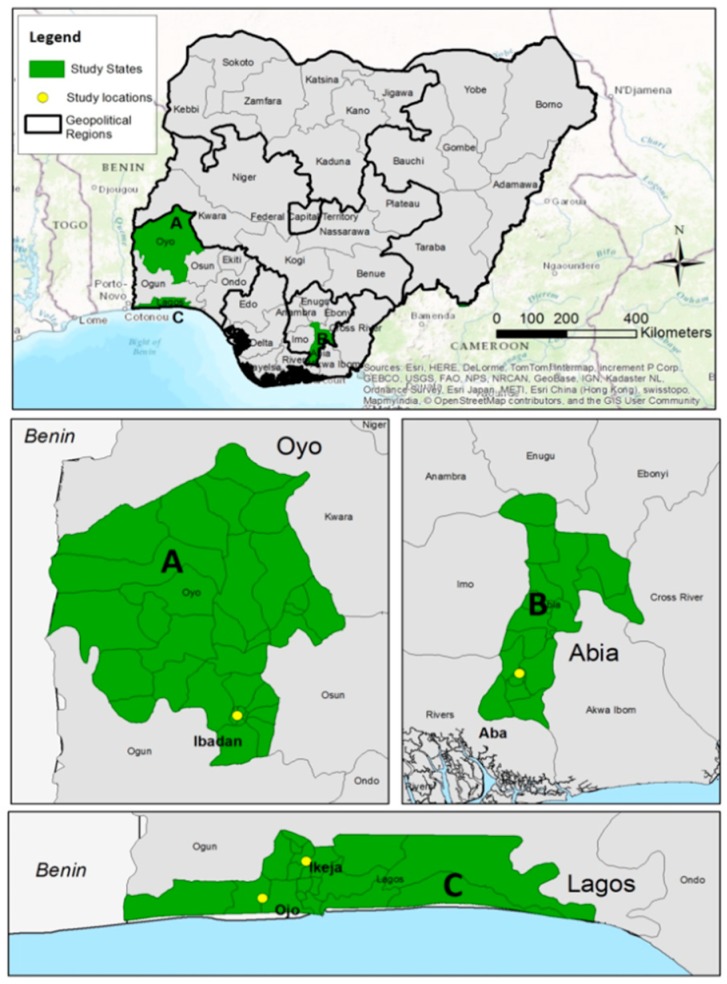
Map of Nigeria showing the study locations.

**Figure 2 ijerph-16-00906-f002:**
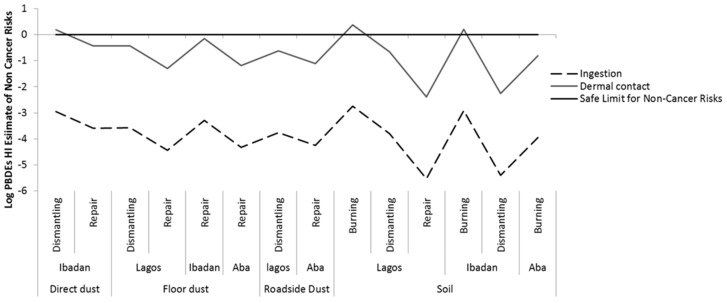
Hazard index (HI) for non-cancer risk via ingestion and dermal contact of Polybrominated Diphenyl Ethers (PBDEs) in soil and dust at various *e*-waste sites in three locations.

**Figure 3 ijerph-16-00906-f003:**
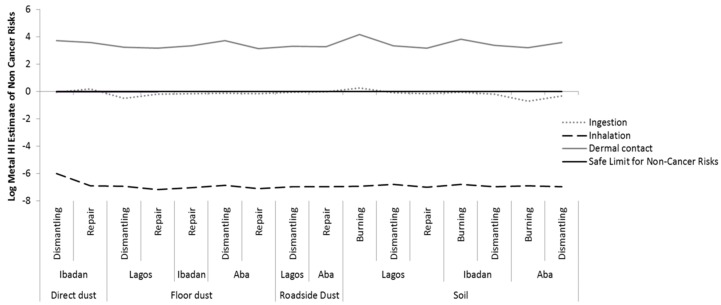
Hazard index (HI) for non-cancer risk of metal exposure via ingestion, inhalation, and dermal contact of soil and dust at various *e*-waste sites in the three locations, showing that *e*-waste workers are prone to non-cancer risks via dermal contact with metals in soils and dust, also via the ingestion of top soils at burning sites in Lagos.

**Figure 4 ijerph-16-00906-f004:**
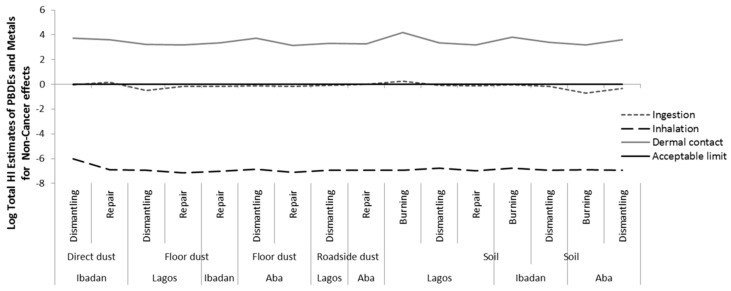
Total Hazard index (HI) for non-cancer risk via each exposure route of metal and Polybrominated Diphenyl Ethers (PBDEs) in soil and dust at various *e*-waste sites in the three locations.

**Figure 5 ijerph-16-00906-f005:**
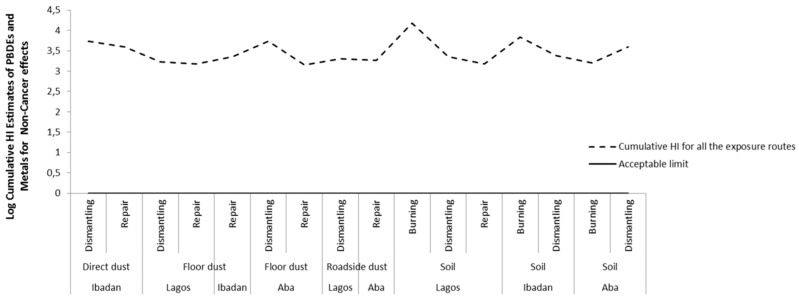
Cumulative Hazard index (HI) of all exposure routes for non-cancer risk of metal and PBDEs in soil and dust at various *e*-waste sites in the three locations.

**Figure 6 ijerph-16-00906-f006:**
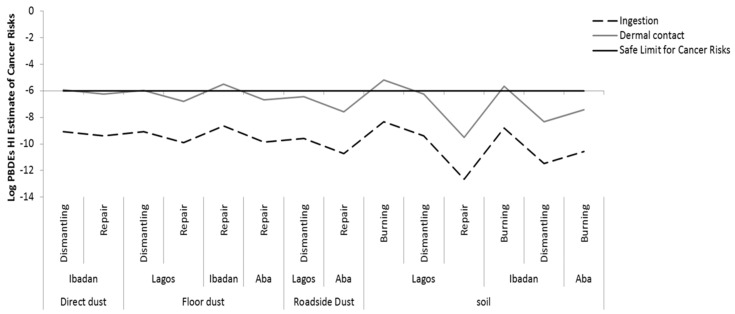
Hazard index (HI) for cancer risk via ingestion and dermal contact of Polybrominated Diphenyl Ethers (PBDEs) soil and dust at various *e*-waste sites in the three locations.

**Figure 7 ijerph-16-00906-f007:**
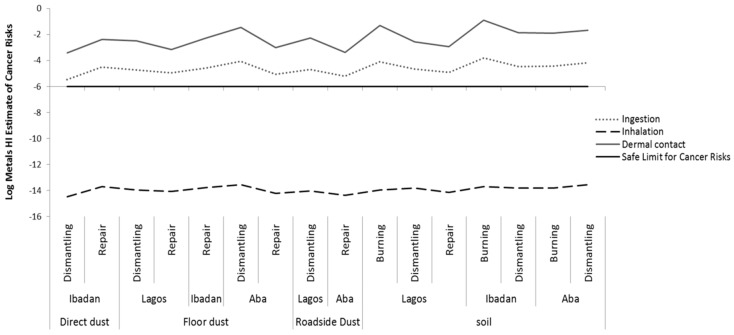
Hazard index (HI) for cancer risk via ingestion and dermal contact of metals in soil and dust at various *e*-waste sites in the three locations.

**Figure 8 ijerph-16-00906-f008:**
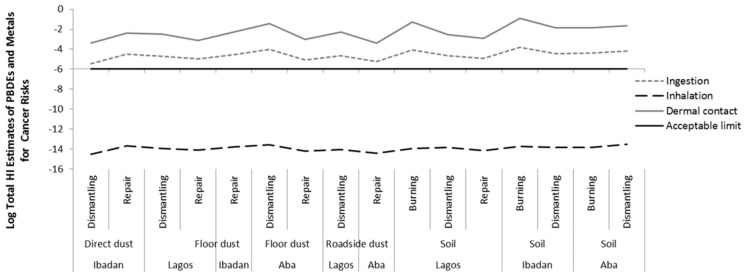
Hazard index (HI) for cancer risk via each exposure route of metal and Polybrominated Diphenyl Ethers (PBDEs) in soil and dust at various *e*-waste sites in the three locations.

**Figure 9 ijerph-16-00906-f009:**
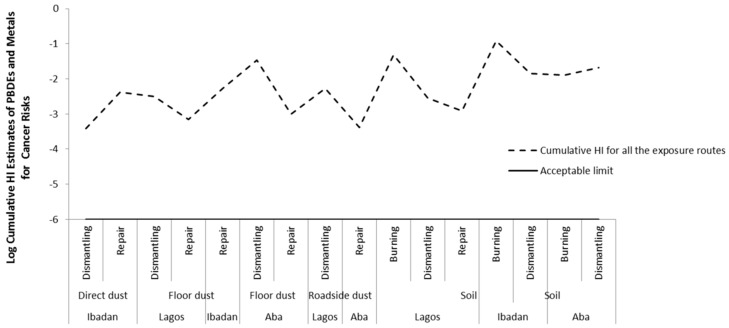
Cumulative Hazard index (HI) of all exposure routes for cancer risk of metal and Polybrominated Diphenyl Ethers (PBDEs) in soil and dust at various *e*-waste sites in the three locations.

**Figure 10 ijerph-16-00906-f010:**
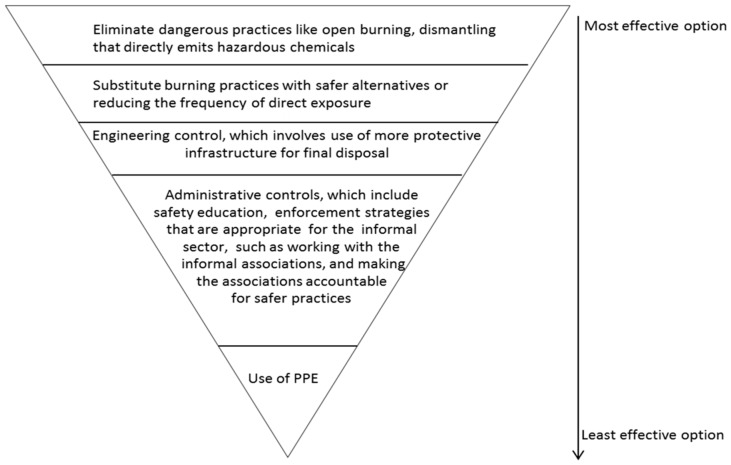
Hierachical control at the informal *e*-waste recycling sites; modified from OSHA 2016, [64].

**Table 1 ijerph-16-00906-t001:** Evidence of health effects due to long-term exposure to persistent organic contaminants.

Chemical	Effects	Reference
PCDD/Fs	Thyroid function	[20]
PBDEs	Thyroid function, Reproductive health, endocrine disruption	[20,21,22,23]
PATHs, PFOA	Reproductive health	[24,25]
Cr, Mn, Ni	Lung function	[26]
Pb, Cr, Cd, Ni	Reproductive health	[27,28,29,30]
PCBs	Reproductive health, thyroid function	[20,29]
Mn, Ni, Pb	Growth	[26,31,32]
Pb	Mental health outcomes	[28,33]
As, Cd, Ni, Cr, Hg, Cu	Cancer, oxidative stress, DNA damage	[32,34]

PCDD/Fs: Polychlorinated Dibenzo-p-dioxins, and Furans; PBDEs: Polybrominated Diphenyl Ethers; PATHs: Polycyclic Aromatic Hydrocarbons; PFOA: perfluorooctanoic acid; PCBs: Polychlorinated Biphenyls.

**Table 2 ijerph-16-00906-t002:** Exposure parameters for adults (*e*-waste workers) with associated references.

Abbreviations	Exposure Factors	Exposure Values	References
C (mg/g)	Median Concentration of the PBDE or metals	Shown in Appendix A	This study
R_ing_ (mg/day)	Ingestion rate	30 mg/day	[47]
R_inh_ (m^3^/day_)_	inhalation rate	20 m^3^/day	[45]
EF (days/year)	Exposure frequency	313 days/year	This study
Work days	Average work days	6 days/week	This study
ED (years)	Exposure duration	24 years	[45]
ET (hours/day)	Exposure time in hours/day at work	9 h/day	This study
BW (kg)	Average body weight (279 workers)	67 kg	This study
AT (days)	Average time (ED × 365 days) for non-carcinogens)	24 × 365 days	[45]
	Average time (70 × 365 days) for carcinogens	70 × 365 days	[45]
Age	Median age of the workers	29 years	This study
SA (cm^2^)	Skin surface area	5700 cm^2^(most of them do not use any PPE)	[48]
AF (unitless)	Skin adherence factor	0.2 mg/cm^2^.day	[45]
ABS (unitless)	Dermal absorption factor	0.1 (for semi-volatile compounds)	[45]
PEF (m^3^/kg)	Particle emission factor	1.36 × 10^9^ m^3^/kg	[45]
CF	Conversion factor	10^−6^	[45]
RfD_i_ (mg/kg/day)	reference dose via ingestion, inhalation, and dermal contact	available for four PBDE congeners and 19 metals	[49]
RfC (mg/m^3^)	Reference concentration	--	[49]
IUR	Inhalation Unit Risk	--	[49]
ADD (mg/kg/day)	average daily dose	Calculated and shown in Appendix A	This study
HQ (unitless)	Hazard quotient	--	
HI	Hazard index	--	
SF	Slope factor	--	[49]

C_dust_: Median Concentration of the PBDE or metals in dust.

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
