# Peer review of "Health Risks of Polybrominated Diphenyl Ethers (PBDEs) and Metals at Informal Electronic Waste Recycling Sites"

_ijerph, 2019, doi:10.3390/ijerph16060906_

Round 1

Reviewer 1 Report

I attached the specific comments for authors.

Author Response

Thank you so much for your time and useful comments. Please see the attached response.

Reviewer 2 Report

This study represents a phenomenal amount of work by the authors, and the extensive supplement with their data really adds value to the manuscript.  The use of the EPA methodology is commendable.  The inclusion of Table 2, the equations, and Supplement Tables 15 & 16 is greatly appreciated.  The subject matter is important, the risks are serious and the addition of recommendations towards the end of the article is helpful. I support publication of this article, with revisions.

Major revisions needed:

1) The abstract does not reflect the scope of the assessment and needs to be substantially rewritten.

2) In the intro, articles # 49, 50 and 51 should be included.  The authors then need to explain how their data/article adds to what has already been published on this topic.

3) Line 111:  Even if the methods have been well detailed in their previous studies, a summary is needed here.  For instance, the reader doesn't even know which instrumentation and methods were used to analyze metals and organics concentrations, nor where that work was done.  Lines 122-126, it isn't clear if 62 samples for metals analysis were collected from each site, or across all sites?  The data in the supplement give median values, but no standard deviations nor "n" values.  This needs clarification.  Line 327-329: this should be in the methods section, and deserves more explanation than provided here.  If it was previously published, the citation should be provided.  Lines 321-326 might also be moved to the methods?

4) Lines 127-128:  "The difference in the number of samples analyzed for metals and PBDE is because... some samples were beyond detection limits" - First of all, do the authors mean below the detection limits?  More importantly, does this mean that values below the detection limits were thrown out of the analysis?  If so, the entire analysis needs to be re-done.

5) Line 302-303:  "We also considered it appropriate to sum risks from multiple exposure routes..." (Figures 5 and 9).  Please cite specific pages in EPA guidance which validate doing so.

6) Table 2:  Why is the exposure duration for carcinogenic risks 70 years??

7) Line 212 states that additional risk in the range of 10E-6 to 10E-4 is "acceptable," but lines 283 and 293 states that added risks > than 10E-6 "are above the safe limit."  Be consistent.

Other suggestions

Line 76:  How is "noise" an "effect of exposure?"

Lines 82-83:  Awkward, rephrase

L 94: eliminate "to"

L 97:  "floor dust" is duplicated

L 112:  substitute "of" for "on"

L 141:  substitute "precautions" for "caution"

L 153: remove "that"

L 249-250:  Saying "particularly for metals," implies that this is also true but to a lesser extent for PBDEs, but that isn't the case.  Reword to avoid giving this impression

L255:  it would be more accurate to say, "Generally for the four PBDEs for which RfDs were available..." 

L282:  suggest adding "depending on the site" to the end of this sentence. 

L283:  must qualify that this is in some locations only, or give a percentage of sites which exceeded the safe limit. 

L371-372:  awkward, reword.

L408 + 414:  incomplete sentences.

There are some other minor grammatical issues which need to be addressed.

L 428-429:  "and those in the new dutch list"  - neither grammatical nor clear what is meant by "new dutch list."

L434-435:  The study does not estimate risks for people in the vicinity of recycling.  There is no evidence that "people in the vicinity" are working in these dusts 6 days a week, 9 hours/day, for 24 years. 

Author Response

(The authors gave the same response as above.)

Reviewer 3 Report

This manuscript is well written the results are clearly presented. I appreciated the inclusion of the data as supplementary material. I do think that it is very interesting that inhalation was found to be nominally related to health risk, especially considering that some sites were associated with burning. Might this be related to the reliance on soil and dust samples? My point is that ultra fine particles are likely to stay airborne and would not be represented in the samples taken. It is worth mentioning that a lack of air samples is a limitation with the current approach and that inhalation may actually be a significant route as well.

Author Response

(The authors gave the same response as above.)

Round 2

Reviewer 1 Report

Minor suggestion

L289: Add "in Ibadan" after "~at repair sites".

L397: I couldn't find supplementary figures 1d-f in revision supplementary information. Maybe omit. 

In all text: Uniform a font of characters of "e-waste" or "e-waste". 

Supplementary table 7~12: Not mention in the text. Should be described the estimated ADD in section 3.1. According to this revision, authors should be re-numbered of supplementary tables. 

Supplementary table 8, 10 and 16: In the caption, should be written "Supplementary table" same as other tables.

Supplementary table 13: ing, der and inh should be subscript.

Author Response

Reviewer 1

Open Review

English language and style

( ) Extensive editing of English language and style required 
( ) Moderate English changes required 
(x) English language and style are fine/minor spell check required 
( ) I don't feel qualified to judge about the English language and style 

Yes

Can be improved

Must be improved

Not applicable

Does the introduction provide sufficient background and include   all relevant references?

(x)

( )

( )

( )

Is the research design appropriate?

(x)

( )

( )

( )

Are the methods adequately described?

(x)

( )

( )

( )

Are the results clearly presented?

(x)

( )

( )

( )

Are the conclusions supported by the results?

(x)

( )

( )

( )

Comments and Suggestions for Authors

Minor suggestion

L289: Add "in Ibadan" after "~at repair sites".

Reply: Thank you for this, “in Ibadan” has been added, see line 290 in the clean copy

L397: I couldn't find supplementary figures 1d-f in revision supplementary information. Maybe omit. 

Reply: Thank you for this observation, it has been added as part of the supplementary information

In all text: Uniform a font of characters of "e-waste" or "e-waste". 

Reply: e-waste has been replaced with e-waste

Supplementary table 7~12: Not mention in the text. Should be described the estimated ADD in section 3.1. According to this revision, authors should be re-numbered of supplementary tables. 

Reply: Supplementary tables 7-12 were mentioned in table 2, line 207 in the clean copy

Supplementary table 8, 10 and 16: In the caption, should be written "Supplementary table" same as other tables.

Reply: Thank you, “supplementary” is now written in fullin all the mentioned tables

Supplementary table 13: ing, der and inh should be subscript.

Reply: This has been adjusted. Thank you

Submission Date

21 December 2018

Date of this review

25 Feb 2019 13:47:11

Reviewer 2 Report

This article is much improved, and I appreciate all the hard work the authors have put into addressing the reviews, and their detailed responses to my suggestions.  There are still a handful of points I would like to see addressed, and overall English language editing is still needed, but it is close to publishable.  This is an important, substantive study and I commend the authors on their work.

My primary concern is that if the authors do not have adequate air quality data to robustly analyze the inhalation exposures, it would be far better to take the inhalation calculations out of the article and simply say the inhalation risks urgently need to be analyzed.  Underestimating the inhalation risks due to inadequate data could create false reassurance for e-waste workers, others working and living in the vicinity, health professionals, regulators  and the e-waste industry– and hence could work against anything being done to control emissions from burning, for instance.

 *** I have copied and pasted sections from the authors' response into this document, but see that they did not paste into this.  I will try to attach the complete document here, and will also email to Editor Tao.

 Yes, the rewrite has really improved this, thank you.  I would still recommend rewording this sentence: “The HI values for dermal exposure to soil and dust were greater than one at all e-waste recycling sites (burning, dismantling, and repair sites), particularly for metals.” This implies that the HI values for dermal exposure were greater than one for PBDEs as well as for metals, which isn’t the case. It would be better to say:  The HI values for dermal exposure to metals and PBDEs combined, via soil and dust, were greater than one at all e-waste recycling sites (burning, dismantling, and repair sites), with metals contributing the great majority of the risk.

Note:  The copy I have is the one that shows the edits, hence the line numbers are from that version of the article:

Line 270 should read “higher than acceptable probability…”

Line 275 - slope factor (SF) or the Inhalation Unit Risk (IUR)  [not suggesting the word be underlined, just indicating what I would change]

Line 297 – Citation is lacking here:  “we estimated the total non-cancer hazard by summing up the HIs of metals and PBDEs for each of the exposure routes.”  Also, comparable statement on line 343. Line 375 finally gives this citation:  “We also considered it appropriate to sum risks from multiple exposure routes (i.e the cumulative risk of exposure) [49].” 

First, the citation should be given the first time this cumulative risk assessment method is mentioned.  More importantly, reference #49 is not the reference for this given in the authors’ response to the initial review:    

This is critical to correct.  Also, if this citation is misnumbered, the other citations should be checked for accurate numbering.

Line 360 +364:  Here the authors are stating that the added risks via ingestion (in the 10-6 to 10-4 range) are “unacceptable,” whereas higher up in the article it is noted that added risks in this range are “within the acceptable limits”.  Be consistent. 

Line 401”: …epidemiological methods to select the target groups and sites to ensure that the results obtained are a representative of the target groups and sites…” please add citation(s).

Line 481   ” …important to note that non-e-waste workers, residents, and children around the e-waste sites are equally at risk of adverse health effects from informal e-waste recycling as also pointed out  by[57,60,15,56]. The other informal non-e-waste workers around the e-waste recycling vicinity have similar socio-demographics and work conditions [61].  Thank you for the added citations, that is helpful.  Nevertheless, it is a huge claim to say they are “equally” at risk as they aren’t handling all the same materials, doing exactly the same activities, and children might be even more at risk.  It would be fair to say they are very likely to also be at risk.

The grammar has improved significantly with the edits.  There are still minor grammatical errors in English throughout the article which should be corrected by an editor.  For instance, line 231 – potential health risks to workers were…, not “risk of workers.”  Line 221-222:  sites …turn into… communities.   Line 216 – released.  Line 251-252 – tacking the clause on the end of the sentence is awkward.  Line 261 should be “non-cancer.” In the “We recommend” section, each of the items listed below should employ parallel grammar.  These are just by way of example – I cannot possibly list them all. 

Author Response

Reviewer 2

Open Review

English language and style

( ) Extensive editing of English language and style required 
(x) Moderate English changes required 
( ) English language and style are fine/minor spell check required 
( ) I don't feel qualified to judge about the English language and style 

Comments and Suggestions for Authors

This article is much improved, and I appreciate all the hard work the authors have put into addressing the reviews, and their detailed responses to my suggestions.  There are still a handful of points I would like to see addressed, and overall English language editing is still needed, but it is close to publishable.  This is an important, substantive study and I commend the authors on their work.

My primary concern is that if the authors do not have adequate air quality data to robustly analyze the inhalation exposures, it would be far better to take the inhalation calculations out of the article and simply say the inhalation risks urgently need to be analyzed.  Underestimating the inhalation risks due to inadequate data could create false reassurance for e-waste workers, others working and living in the vicinity, health professionals, regulators  and the e-waste industry– and hence could work against anything being done to control emissions from burning, for instance.

 *** I have copied and pasted sections from the authors' response into this document, but see that they did not paste into this.  I will try to attach the complete document here, and will also email to Editor Tao.

 Yes, the rewrite has really improved this, thank you.  I would still recommend rewording this sentence: â€śThe HI values for dermal exposure to soil and dust were greater than one at all e-waste recycling sites (burning, dismantling, and repair sites), particularly for metals.” This implies that the HI values for dermal exposure were greater than one for PBDEs as well as for metals, which isn’t the case. It would be better to say:  The HI values for dermal exposure to metals and PBDEs combined, via soil and dust, were greater than one at all e-waste recycling sites (burning, dismantling, and repair sites), with metals contributing the great majority of the risk.

Reply:  Thank you, this has been re-worded

Note:  The copy I have is the one that shows the edits, hence the line numbers are from that version of the article:

Line 270 should read “higher than acceptable probability…”

Reply: Thank you, Reworded. See line 220.

Line 275 - slope factor (SF) or the Inhalation Unit Risk (IUR)  [not suggesting the word be underlined, just indicating what I would change]

Reply: Reworded. See line 225

Line 297 – Citation is lacking here:  “we estimated the total non-cancer hazard by summing up the HIs of metals and PBDEs for each of the exposure routes.”  Also, comparable statement on line 343. Line 375 finally gives this citation:  “We also considered it appropriate to sum risks from multiple exposure routes (i.e the cumulative risk of exposure) [49].” 

Reply: The reference has also been inserted, see Line 246 in the clean copy.

First, the citation should be given the first time this cumulative risk assessment method is mentioned.  More importantly, reference #49 is not the reference for this given in the authors’ response to the initial review:    

This is critical to correct.  Also, if this citation is misnumbered, the other citations should be checked for accurate numbering.

Reply: Thank you for this observation: The reference is present in the clean copy and we verified that it has the proper number, which was indeed the case. The renumbering was already done in the previous version of the clean copy.

 Line 360 +364:  Here the authors are stating that the added risks via ingestion (in the 10-6 to 10-4range) are “unacceptable,” whereas higher up in the article it is noted that added risks in this range are “within the acceptable limits”.  Be consistent. 

Reply: Apologies for not reflecting this correction the edited version, the clean copy contains the accurate figures, see line 309 in the clean copy

Line 401”: …epidemiological methods to select the target groups and sites to ensure that the results obtained are a representative of the target groups and sites…” please add citation(s).

Reply: There is no specific reference. The main author has a background in public health, so the background knowledge was employed  which  strengthened the sample design. This ensured adequate representation of the sites and the target population.

 Line 481   ” …important to note that non-e-waste workers, residents, and children around the e-waste sites are equally at risk of adverse health effects from informal e-waste recycling as also pointed out  by[57,60,15,56]. The other informal non-e-waste workers around the e-waste recycling vicinity have similar socio-demographics and work conditions [61].  Thank you for the added citations, that is helpful.  Nevertheless, it is a huge claim to say they are “equally” at risk as they aren’t handling all the same materials, doing exactly the same activities, and children might be even more at risk.  It would be fair to say they are very likely to also be at risk.

Reply: Thank you for pointing this out, the sentence has been re-worded, see line 424 in the clean copy

 The grammar has improved significantly with the edits.  There are still minor grammatical errors in English throughout the article which should be corrected by an editor.  For instance:

line 231 – potential health risks to workers were…, not “risk of workers.” 

Reply: Thank you, this was corrected in the clean copy, see line 180 in the clean copy

Line 221-222:  sites …turn into… communities.  

Reply: rephrased accordingly, see line 174 in the clean copy.

Line 216 – released. 

Reply: The sentence has been reworded, see line 168 in the clean copy

Line 251-252 – tacking the clause on the end of the sentence is awkward. 

Reply: The sentence has been reworded, see lines 198 -199 in the clean copy

Line 261 should be “non-cancer.”

Reply: This has been corrected, see line 208 in the clean copy

In the “We recommend” section, each of the items listed below should employ parallel grammar.  These are just by way of example – I cannot possibly list them all. 

Reply: Thank you, we have re-read the part on 'we recommend' to make sure there are no grammatical errors, see line 437-438
